# Attitudes of Polish Medical and Health Sciences Students towards Persons with Physical Disabilities Using the MAS-PL Scale

**DOI:** 10.3390/ijerph18157787

**Published:** 2021-07-22

**Authors:** Iwona Radlińska, Marta Kożybska, Beata Karakiewicz

**Affiliations:** 1Subdepartment of Medical Law, Department of Social Medicine, Faculty of Health Sciences, Pomeranian Medical University in Szczecin, ul. Żołnierska 48, 71-210 Szczecin, Poland; iwona.radlinska@pum.edu.pl; 2Subdepartment of Social Medicine and Public Health, Department of Social Medicine, Pomeranian Medical University in Szczecin, Żołnierska Str. 48, 71-210 Szczecin, Poland; beata.karakiewicz@pum.edu.pl

**Keywords:** disability, medical students, multidimensional attitudes scale towards persons with disabilities, MAS scale

## Abstract

Discovering the role of negative attitudes in the social functioning of people with disabilities, tools were developed to measure these attitudes, of which the Multidimensional Attitude Scale of People with Disabilities (MAS) is a good one. It is particularly important to study the attitudes of people who are professionally involved in meeting the needs of people with disabilities. The aim of this study was to determine the attitudes towards people with physical disabilities among medical and health sciences students regarding gender, year of study, field of study, and place of residence. The study was conducted among 625 students in Poland with the use of the MAS-PL scale. The results obtained indicate that women display more positive attitudes towards people with disabilities than men in the cognition and behavioural domains. In the emotional domain, these attitudes were more negative than in the other domains and almost identical for both genders. The year of study, field of study, and place of residence did not differentiate students in terms of their MAS score. Since the sociodemographic variables studied do not determine a positive attitude, educational interventions to increase contact with people with disabilities should be undertaken in future research, and this factor should be investigated as an element of attitude modulation.

## 1. Introduction

### 1.1. Concept of Disability and Attitudes towards People with Disabilities

In defining disability, we would like to refer to the definition from the UN Convention on the Rights of Persons with Disabilities (CRPD), which fully reflects the social model of disability. In this view, a person with a disability appears as a subject and a full human being and not as an object of care. According to the CRPD [1], a person with a disability is a person who has a range of psycho-physical limitations (long-term physical, mental, intellectual or sensory impairments) “which in interaction with various barriers may hinder their full and effective participation in society on an equal basis with others” (Art. 1 CRPD). 

It is emphasised that impairment alone is not a limitation to participation, but is in fact a barrier. Physical barriers (lack of wheelchair ramps, inadequate pedestrian crossings, etc.) are easy to overcome with adequate resources. More problematic, however, are the barriers inherent in the mentality of societies, which lead to the stigmatisation and social exclusion of people with disabilities. Unfortunately, these negative attitudes have been rooted in societies for a long time [2,3]. 

The issue of attitudes is a field of social psychology that is variously defined [4]. In the structural, traditional view, the concept of attitude consists of three components: a cognitive component (knowledge and beliefs), an affective component (emotional-motivational), and a behavioural component (motivational or action: behavioural intentions) [5]. Attitude is a favourable or unfavourable reaction towards someone or something, expressed in the beliefs, the feelings or the intentional behaviour. The cognitive element of attitude is mainly manifested in categorisation and stereotyping. The affective element means an emotional way of interpreting the environment and leads to the realisation of the behavioural element, i.e., a positive emotion can lead to altruism, and a negative emotion to prejudice and discrimination [4]. It should be added that it is impossible to avoid categorisation, because according to psychologists, it is a method of human perception of reality. However, one of the consequences of categorisation is the use of a stereotype, i.e., a set of specific features that a priori we assign to a given object. A stereotype is characterised by low relevance to the reality (a simplification), social character, and rigidity and permanence.

Prejudice is more problematic because of its more direct relationship with the behaviour and its negative social consequences. Prejudice, like stereotyping, is characterised by inflexibility, irrationality, and overgeneralisation and has the affective and typically negative dimension (it is an unfair and undeserved emotional reaction) [4]. Prejudice is a hostile or negative attitude towards a certain distinguishable/separable group. It can result in a discriminatory behaviour, which involves behaving negatively towards a person based on their membership in a particular group. The causes of prejudice are very diverse and can include both internal problems—personality (including displaced aggression) and external—social problems (conformism to existing social norms or an economic rivalry) [6].

The origins of attitudes are also twofold: they lie in a person’s internal dispositions and their social experiences. At the same time, the social reference of an attitude is very strong. This is crucial when trying to modify attitudes, which are usually difficult to change.

Educational activities and social campaigns are usually used to attempt to influence existing stereotypes or prejudices. One of the indications of the CRPD is to raise public awareness about disability, to promote a positive image of a person with a disability in order to achieve full social inclusion and to prevent discriminatory behaviour.

In this context, it is extremely important to examine the attitudes towards people with disabilities held by those who are professionally involved in meeting their broadly understood needs, which determines their participation in society on equal terms with non-disabled persons. Particularly important for the health of people with disabilities is a positive attitude on the part of those providing health services, including those studying in the medical professions. It seems that as early as at the stage of education, these people should be developed towards openness to the needs of people with disabilities in order to create an adequate therapeutic relationship based on partnership. The positive attitudes of a person working with a person with disabilities are those that allow for the introduction of “support” rather than “care”, are not generated by pity or fear and do not equate a person with a disability with a passive victim of fate. The positive attitude implies an active role for the person with a disability to the best of his or her ability, regardless of the type and severity of the disability, and such involvement should be expected and demanded of the employee [7]. 

### 1.2. Tools for Measuring the Attitudes towards People with Disabilities vs. MAS Scale

While discovering the role of negative attitudes in the social functioning of selected groups, tools to measure these attitudes were developed. To date, many tools have been created to measure the attitudes towards people with disabilities, the best known and longest used of which is the Attitudes Toward Disabled Persons (ATDP) scale created in 1960 by Yuker et al. [8], which measures the affective and behavioural domains. The ATDP scale has become a reference for many other scales created, including the widely used and translated [9,10,11,12,13,14,15,16,17,18,19,20,21,22,23] Multidimensional Attitudes Scale Toward Persons with Disabilities (MAS) [24], which measures attitudes in all three dimensions: thoughts, feelings, and behaviours. The MAS scale can be used as a tool to measure attitudes in a variety of settings, including after the application of an educational intervention and in relation to people with different types of disabilities by modifying the scenario of the illustration that begins the survey. Details of the creation and validity of the Polish adaptation of the MAS-PL scale used as our research tool have already been published [19].

Of course, in addition to these scales, there are dozens of instruments to measure the attitudes used appropriately to the types of disabilities or social groups [25,26], such as the recently developed WHO Attitudes to Disability Scale (WHO ADS) [27], which has been used successfully in various populations [28]. Although the WHO ADS is interesting in distinguishing attitudes to discrimination and inclusion, it does not include an affective component that reveals the emotional attitudes and thus the sincerity of the stated beliefs and behaviours [29]. In addition, numerous studies have demonstrated the relationship between negative attitudes towards people with disabilities and the affective component—especially anxiety [30,31] and aggression [32]. Thus, we appreciated the MAS scale for including all three attitude components (multidimensional scale), which are recommended in the development of attitude measurement tools [25]. In addition, the MAS scale includes a projection mechanism by introducing the respondents into a scenario in which they participate as a “narrator” and are asked to guess the attitudes of the “main character”. This hypothetical social situation minimises misrepresentation in reporting expected and socially acceptable responses by the respondents [24] and thus may help to reduce the impact of “social desirability” bias on the responses [14]. Thus, it appears that the MAS scale is an ideal multidimensional tool for measuring the attitudes to disability.

### 1.3. Research on Attitudes towards People with Disabilities in Poland in the Context of Global Studies

So far, research on attitudes towards people with disabilities has been conducted in Poland using original research tools but not published in English, and has therefore not been taken into account in systematic reviews. An accurate and reliable Polish tool is primarily the Scale of Attitudes Towards Persons with Disabilities (SATPD) by Sękowski [33]. It is a Likert-type scale consisting of 30 statements (15 positive and 15 negative) about people with disabilities which are similar to or the same as in well-known scales such as the ATDP scale and others [8,34,35].

Professor Sękowski, as a psychologist, has repeatedly used his SATPD scale to assess the general relationships of attitudes to personality or other independent variables. He has demonstrated that there are specific personality traits and their configurations that foster positive or negative attitudes towards people with disabilities. He showed, similar to Siller and Chipman and Yuker et al. [35,36], that high general self-esteem and self-acceptance are good predictors of positive attitudes towards people with disabilities [32]. In contrast, a lower intellectual level is associated with negative attitudes [32], confirming the findings of earlier studies by Whiteman and Lukoff [37]. In addition, he conducted pioneering research into the link between attitudes and value hierarchy, showing that people who exhibit moral values (high evaluation of love, kindness, knowledge, truth, honesty, conscience, and acting in accordance with socio-moral norms) have more positive attitudes towards people with disabilities than those who exhibit “prestige” values (valuing power, career, leadership) [32]. He noted, following other psychologists, the important role of direct contact with people with disabilities, which most often modifies attitudes in a more positive direction [32]. 

In representative Polish studies conducted in the years 1978, 1993, 2000, and 2007 by governmental, statistical research institutions—the Public Opinion Research Centre (pol. OBOP), and the Centre for Public Opinion Research (pol. CBOS) [38,39,40]—the same questions were systematically asked, which allows the dynamics of changes in these attitudes to be tracked. The research in 1978 and 1993 was commissioned by Ostrowska [38]. Until the year 2000, a systematic increase in positive attitudes and a decrease in negative attitudes were observed [39]. On the other hand, comparing the studies of 2000 and 2007, only a 1 percent change in opinions is visible, so opinions have hardly changed. According to the latest survey in 2007, almost half of the Poles surveyed (48%—an increase of 1 point) believed that most of Polish society has negative attitudes towards people with disabilities. The remaining people (45%—down by 1 point) had the opposite opinion [39,40]. This shows that as much as half of the Polish society has negative attitudes. On the other hand, unrepresented studies on the attitudes of young people in Poland conducted after 2000 indicate that a small percentage of them (approximately 10%) have positive attitudes [41,42]. In a study by Nowak [43] published in 2015 on a group of 249 students (attitudes in three dimensions using the author’s tool), the attitudes were overwhelmingly positive, but only in the cognitive dimension (80%). The author also observed a clear increase in negative attitudes with age. High school students showed the most negative emotions and a lot of negative beliefs. As stated by the author, “[t]he overall positive attitude of young people towards people with physical disabilities seems to be largely only declarative”. This indicates the universality of multidimensional tools, which also in Polish studies indicated the full spectrum of attitudes towards people with disabilities.

The aim of this study was to determine, using the MAS-PL scale:(1)Attitudes of medical students towards disabled people;(2)Differences in the attitudes towards disabled people among medical students depending on gender, year, major, and place of residence.

## 2. Materials and Methods

### 2.1. Data Collection and Research Sample

The cross-sectional study was conducted in the academic year 2017/2018 (from October 2017 to June 2018) among students of the medical and health sciences faculties in Pomeranian Medical University in Szczecin. The surveys were collected during regular classes with students by the paper and pencil interviewing (PAPI) technique. Students independently read and completed the questionnaire in class—it was an auditorium questionnaire. The survey included all groups of students of all fields of study within health sciences and dentistry, in courses taught by the teachers of the Department of Public Health and Social Medicine and the Department of Conservative Dentistry. Thus, students at the years of study other than those listed in Table 1 and of other majors of study, such as medicine, pharmacy, and medical analysis, did not participate in this survey. The participation in a survey was voluntary, and some students refused to participate in the survey. The predominance of the female students is specific to medical and health sciences faculties. The research was anonymous and the participants were not remunerated for completing the questionnaire. The study sample consisted of 625 students. 

### 2.2. Measures

The research used the Polish adaptation of the multidimensional attitudes scale towards persons with disabilities (MAS-PL) and a self-designed survey on sociodemographic variables (gender, age, education, place of residence, major of study, year of study).

The MAS-PL is a reliable instrument for studying the attitudes of Poles towards people with disabilities. The MAS-PL scale begins with a social scenario vignette: a description of a scenario of an accidental and forced by circumstances meeting in a cafe of a fully-abled person (Ewa/Adam) with a person in a wheelchair (male/female). The respondent is to imagine this situation and indicate the variety of emotions (list of 16 emotions), thoughts (list of 10 thoughts), and potential behaviours (list of 8 behaviours) it can elicit in non-disabled people. Therefore, the questions are not addressed to the respondent directly, but based on a projection mechanism to have respondents transfer their own emotions, thoughts, and behaviours onto the given situation. The scale measures three components of attitudes: the emotional, cognitive, and behavioural domains. The answers are provided on a 5-point Likert scale, where 1 means “not at all” and 5 “very much”. Higher scores represent more negative attitudes and positive items require reverse scoring. The Polish questionnaire and its psychometric properties were described in more detail in our previous open-access article [19].

### 2.3. Methods of Data Analysis 

The analysis was performed in the licensed Statistica 13.1 package. The Kolmogorov–Smirnov test was used to assess the normality of the distribution of quantitative traits, the non-parametric z test for two independent groups, Spearman’s rank correlation, and the Kruskal–Wallis test for many independent groups. Regression analysis was also performed. The level of significance was set at *α* = 0.05. 

## 3. Results

The sample consisted of 25 students. There were 82.40% (*n* = 515) women and 17.60% (*n* = 110) men in the sample, with a mean age of 23.24 years (*SD* = 3.35). Half of the participants (54.89%; *n* = 348) lived in a city and half in a small town (28.86%; *n* = 183) or a rural area (16.32%; *n* = 102). The full sociodemographic characteristics of the group are presented in Table 1.

Table 2 shows the basic distribution of results. No normal distribution was obtained in either the affective component, the cognitive component, or the behavioural component. The global score is at the borderline of significance. For the global score, the maximum score obtained is much lower than possible. This means that the study participants did not receive the highest possible results. The affective, cognitive and behavioural subscales are characterised by a slight rightness, which suggests the advantage of low scores over high scores. This is additional evidence to conclude that the intensity of negative attitudes in the studied sample is rather low. For the global score and the affective component and the behavioural component, kurtosis suggests a weaker concentration of results around the mean and greater differentiation. For the cognitive scale, a strong concentration of results around the mean and a small differentiation of responses can be observed.

It was shown that men obtained statistically significantly higher results than women in the MAS total score (*p* < 0.01) and in the cognitive (*p* < 0.001) and behavioural (*p* < 0.01) subscales (Table 3). The strength of the effect is higher for the cognitive component, which means a stronger gender relationship with this area. The affective component does not differ significantly between men and women.

As shown in Table 4 below, there were no statistically significant differences between the place of residence and the MAS-PL score (total and subscales). 

Table 5 shows the results of MAS-PL by major of study. There were no statistically significant differences between the major of study and the global score (*H*(8, *N* = 625) = 7.714; *p* = 0.462), the affective component (*H*(8, *N*= 625) = 8.847; *p* = 0.355), the cognitive component (*H*(8, *N* = 625) = 10.611; *p* = 0.225), and the behavioural component (*H*(8, *N* = 625) = 11.449; *p* = 0.178). 

There were no statistically significant correlations between the year of study and age and the MAS-PL score (total and subscales). The results of the analysis are presented in Table 6.

There were statistically significant positive correlations between all components (affective, cognitive, behavioural) and between the components and the total MAS-PL score (Table 7).

The analysis showed that the models had little possibility of predicting the dependent variables (Table 8). They can explain 0 to 4% of the variability of the dependent variables. The developed models predicting the result of the global score, affective and behavioural, are poorly fitted to the data. In predicting the behavioural and global scores, only gender turned out to be a significant predictor. Gender relationships with the global score and behavioural score are negative and weak. This confirms the conclusion that for the indicated areas, men will achieve higher results. In predicting the affective score, only the year of study turned out to be a significant predictor. This relationship is positive and weak. This indicates that in the affective component, students of higher years will receive more points.

The cognitive prediction model turned out to be a good fit for the data, but still explains only 4% of the variability of the dependent variable. In this model, all analysed variables turned out to be significant predictors. The relationship between age and place of residence and the cognitive score is positive and weak, which proves that people of higher age and living in larger towns will score more points in this domain. The relationship between sex and year of study is negative and weak. This means that men and lower-year students will receive higher points in this domain.

Removal of irrelevant predictors from the models in the regression analysis by the forward selection method improved the fit of the model to the data, but did not increase the percentage of the explained variance of the dependent variable. Moreover, the removal of irrelevant predictors from the models did not change the value of the relationship of significant predictors with the dependent variables.

## 4. Discussion

The research results indicate that the attitudes of women towards people with disabilities were more positive than those of men in the cognitive and behavioural domains. Meanwhile, in the emotional domain, these attitudes were more negative than in the other two domains and almost identical for both genders. The year of study, field of study and place of residence did not differentiate students in terms of the MAS score. 

The cognitive domain prediction model turned out to be a good fit for the data. Significant predictors turned out to be age, gender, year of study, and place of residence. People of higher age, people living in larger towns, men, students of lower years will receive more points, i.e., they will have more negative attitudes towards people with disabilities. It should be emphasised that this model explains only 4% of the variability of the cognitive score.

There are two interesting, systematic reviews on student attitudes [44,45]. The first one from 2012 looked, among other things, at medical students’ attitudes towards people with physical disabilities as measured by recognised attitude scales, primarily the ATDP scale. The findings from the review showed that the students had higher levels of attitudes than the general population and the female students had more positive attitudes than male students. This is consistent with the results of our own research. It was also shown that more experience working with people with disabilities, as well as social contact, generated more positive attitudes [44].

### 4.1. Attitudes of Students Measured by Different Research Scales—Field of Study, Gender, Contact

Previous studies have attempted to find differences in the attitudes of students in different fields of study [44]. Attention is drawn to the results of an ATDP scale study from 1991 by Lyons [46], in which the attitudes of occupational therapy students and business majors were compared. This study did not reveal statistically significant differences in the students’ attitudes, even taking into account the year of graduation, as in our own study. However, higher attitudes were shown by students who had personal contact with people with disabilities beyond the context of a caregiver–care receiver relationship. 

The attitudes of rehabilitation and business students were also the subject of a subsequent Chinese study published in 2002 by Chan et al. [47] using the ATDP scale, which showed a significant influence of the college curriculum on the attitudes of students in later years of study. In the first year of the study, rehabilitation students showed more positive attitudes, while in the third year, business students also obtained positive attitudes.

Significant differences in the attitudes of occupational therapy and physiotherapy students were noted by Stachura and Graven in a 2007 study of UK students using the Interaction with Disabled Persons (IDP) scale [48]. The occupational therapy students had significantly more positive attitudes than the physiotherapy students and the students who had informal contact with disabled people (outside of their studies) regardless of the field of study. Work experience with people with disabilities had an impact on increasing positive attitudes of physiotherapy students (better attitudes in higher years of study), but not occupational therapy students (thought to be due to social contact with people with disabilities outside of study) [48].

In our study, there was no evidence that the students of higher years showed more positive attitudes towards disabled persons than the students of lower years. Interestingly, in our study, we checked attitudes towards disabled people in a group of students representing very different medical and health sciences (e.g., administration and management in healthcare, biotechnology, physiotherapy, dentistry). There was no evidence that the level of attitudes differed depending on the field of study. This can be explained in two ways. Firstly, it may be that students of medical and health sciences, as they are oriented towards working with other people, manifest similar attitudes regardless of whether in the future they want to be employees of administration or laboratories or provide direct help to patients. Secondly, there are factors other than the choice of educational path which are related to attitudes towards people with disabilities. The research results of other authors presented above indicate that these are mainly their own personal experiences with people with disabilities. 

Furthermore, studies of medical students conducted with the ATDP scale in the USA and Canada [49], as well as Turkey [50], among others, have shown the same trends—generally higher levels of student attitudes in relation to the general population and lower levels of male student attitudes. 

The second systematic review concerns research (2012–2019) on minor students’ attitudes towards disability and inclusive education [45]. The results of the review confirm the trend found in adults—that is, female students express more positive attitudes towards disability. In general, the students showed more positive attitudes towards overt disability, which is also the rule in adult attitudes. It seems, therefore, that schoolchildren have exactly the same attitudes as the society in which they grow up. 

### 4.2. Attitudes of Students Measured by the MAS Scale—Gender, Contact 

The MAS scale research to date has mostly focussed on students’ opinions, with few exceptions. The studies of MAS in relation to gender are inconclusive. The authors of the MAS [29] have shown that both women and men (general population) show more positive beliefs and thoughts and less distancing behaviour towards wheelchair users than towards non-disabled people (men had slightly worse attitudes in this respect). However, this overall positive result is disturbed by the analysis of the emotions themselves, as both genders also showed negative emotions towards the physically disabled. According to the authors, there may have been an unconscious defence mechanism in the respondents (following Freud [51]), which was realised by trying to hide a socially illegitimate impulse and declaring, often exaggeratedly, acceptable reactions [29]. Moreover, in our study, despite generally more positive attitudes, women also showed negative emotions. An analysis of emotions thus reveals a hidden space (in terms of emotions) and thus the full spectrum of attitudes. 

The lack of a significant gender relationship regarding attitudes has been shown in several MAS scale studies (e.g., [12,14,18,23]), while other relationships have been shown. In a study of German underage students on attitudes towards physical disability, gender was not significant and less than 1/3 of all students had positive attitudes [18]. A Dutch study investigated attitudes towards people who were deaf, blind, paralysed, or intellectually disabled. Higher age and familiarity with the disabled person had a significant positive effect on attitudes, while self-esteem and gender had only a minor effect [12]. In Korean students’ banshees (visible disability), non-significant results were revealed for gender and even previous contact, and highlighted the role of “social desirability” in shaping attitudes (the research was also conducted with the Marlowe–Crowne Social Desirability Scale: MC-SDS) [14].

In contrast, other authors who examined attitudes using an adaptation of the MAS scale showed significantly more positive attitudes among women than men—but usually in selected attitude subscales. The results from a Serbian study towards people with physical disabilities indicated more negative emotions and avoidance behaviour among the men, and more both positive and perceptual cognitions and approaching behaviour among the women. However, the men were found to be more likely than women to experience deep-seated negative emotions such as disgust, indifference and the feeling of guilt [10]. In contrast, in a study of Ethiopian students on visible disability, both gender and year of study were significant only for the cognition subscale, and the type of major studied was significant on both the affect and behavioural subscales. A strong correlation was found between self-esteem and attitudes towards disability. Those with high self-esteem also had more positive attitudes. Ethiopian college students have negative attitudes in general based on this measure [11]. French studies have shown significant results towards individuals with autism in the cognitive domain (more positive cognitive attitudes) [16].

In general, MAS scale studies note that females possessed more positive cognitive and behavioural attitudes but more negative emotions than men, e.g., in French and German studies [16,18]. Perhaps this tendency is related to women’s acceptance of the socially assigned role of women as mothers or caregivers (perhaps even on a biological, subconscious level) [29].

### 4.3. Limitations

Results cannot be generalised to the rest of the medical and health sciences student population as they show the attitudes of students from one medical school. 

Attitudes towards people with physical disabilities were measured. Further research is needed to measure attitudes towards other types of disability, in particular towards people with mental disorders.

## 5. Conclusions

Given the inconsistency of results among existing MAS studies, it can be assumed that gender has an effect on the attitudes as measured by MAS depending on nationality and type of disability. The results of our Polish study indicate that women manifested more positive attitudes towards people with physical disabilities than men in the cognitive and behavioural domains. Meanwhile, in the emotional domain, these attitudes were more negative than in the other two domains and almost identical for both genders.

It is stressed that MAS scale scores do not prejudge overall attitudes for two reasons. 

Firstly, these are declared attitudes and despite the use of the MAS scale’s projection mechanism, they may be different from actual attitudes. In view of this, future research with the Social Desirability Scale in combination with the MAS scale is particularly desirable and will show even more about the complexity and ambiguity of declared attitudes.

Secondly, a positive score on even two subscales may create not quite valid results of positive attitudes in general (then the total score for the three subscales may be quite high, but one of them will not score well). This is why tools that measure only two dimensions of attitudes, such as the ATDP scale, appear to be imperfect. A low level in the affect subscale gives a signal that there are deep-rooted negative emotions, or stereotypes. Efforts to hide negative affect show the positive aspect of a person’s behaviour, who wants to behave better than they feel. However, insincerity in this area disrupts the relationship with the disabled person, who senses these excessive efforts and consequently feels negatively about the relationship (sense of inferiority of the disabled person) [29]. This negative affect can be changed by deepening the relationship with the disabled person. All studies, conducted with different tools, indicate that the most positive attitudes are held by people who have social contact with people with disabilities, and this is a more important factor than the others (gender, field or year of study, place of residence). This means that continuous educational interventions, above all enabling the greatest possible contact with people with disabilities, including inclusive education for students with disabilities, are needed to change attitudes in this respect. Various inclusive activities, even unrelated to the field of study, can serve this purpose.

Since the sociodemographic variables studied do not determine a positive attitude, educational interventions to increase contact with people with disabilities should be undertaken in future research and this factor should be investigated as an element of attitude modulation.

## Figures and Tables

**Table 1 ijerph-18-07787-t001:** Sociodemographic characteristics of the sample (N = 625).

Variable		Year of Study of the Studied Group	Study Group M/N	Study Group SD/%	Number of Students at the University by Gender, Major, and Year of Study	Percentage of the Surveyed Students among All Students by Gender/Major of Study/Year of Study
Age			23.03	3.40		
Gender	male		110	17.0	517	21.47
	female		515	82.40	1489	35.12
Place of residence	rural areas		103	16.25		
	small town		183	28.86		
	city		348	54.89		
Major of study (full-time studies)	administration and management in healthcare	1 year 2-degree	9 (5 women)	1.58	9 (5 women)	62.50 (50.00 women)
		2nd year of 2nd	1 (1 women)		7 (7 women)	
	biotechnology	1 year 2-degree	17 (17 women)	2.72	23 (19 women)	20.43 (20.99 women)
	dietetics	2nd year of the 1st cycle	29 (29 women)	13.76	33 (30 women)	44.33 (47.51 women)
		1 year 2-degree	30 (30 women)		31 (29 women)	
		2nd year of 2nd cycle	27 (27 women)		31 (31 women)	
	physiotherapy	3rd year of the 1st degree	38 (27 women)	22.24	70 (49 women)	40.52 (60.49 women)
		1 year 2-degree	62 (52 women)		68 (53 women)	
		2nd year of 2nd cycle	39 (28 women)		66 (45 women)	
	cosmetology	3rd year of the 1st degree	35 (35 women)	10.88	36 (36 women)	33.99 (33.59 women)
		2nd year of 2nd cycle	34 (34 women)		36 (36 women)	
	nursing	2nd year of the 1st cycle	44 (41 women)	6.94	72 (67 women)	12.09 (12.31 women)
	obstetrics	2nd year of the 1st cycle	2 (2 women)	3.47	25 (25 women)	16.54 (16.54 women)
		1 year 2-degree	20 (20 women)		26 (26 women)	
	emergency medical services	1 year of 1st cycle	18 (3 women)	2.84	18 (7 women)	31.03 (9.09 women)
	dentistry	1 year of long-cycle studies	71 (69 women)	34.70	105 (73 women)	51.76 (54.13 women)
		3rd year of long-cycle studies	91 (39 women)		82 (57 women)	
		5th year of long-cycle studies	58 (56 women)		71 (52 women)	
Year of study	1 year of 1st cycle and 1 year of long-cycle studies		109	17.44	773	14.10
	2nd year of the 1st cycle and 2nd year long-cycle studies		31	4.96	638	4.86
	3rd year of the 1st degree and 3rd year of long-cycle studies		131	20.96	585	22.39
	1 year 2-degree and 4-year long-cycle studies		182	29.12	429	42.42
	2nd year of 2nd cycle and 5th, 6th year of long-cycle studies		172	27.52	620	27.74

M—mean; SD—standard deviation.

**Table 2 ijerph-18-07787-t002:** Descriptive analysis of results (N = 625).

MAS-PL (Total Score)	Range		M	SD	Me	SKE	K	K-S	
	Possible	Gain						d	*p*
Global score	34–170	40–119	80.15	81	14.57	−0.03	−0.41	0.030	0.068
Affective	16–80	16–65	39.12	39	9.08	0.17	−0.47	0.060	0.001
Cognitive	10–50	10–50	23.23	23	6.05	0.39	1.07	0.067	<0.001
Behavioural	8–40	8–35	17.8	18	5.28	0.31	−0.36	0.086	<0.001

M—mean; SD—standard deviation; Me—median; SKE—skewness; K—kurtosis; K-S—Kolmogorov–Smirnov test of normality.

**Table 3 ijerph-18-07787-t003:** Assumption of z tests of the results of MAS-PL scores (total score, affective, cognitive, and behavioural subscales) by gender (N = 625).

MAS-PL (Total Score)	Woman N = 515		Man N = 110		z	*p*	r_g_
	M_rank_	Me	M_rank_	Me			
Global score	304.42	80	353.16	83.5	−2.570	0.010	−0.16
Affective	314.05	39	308.09	39	0.314	0.753	0.02
Cognitive	298.78	22	379.6	26	−4.261	<0.001	−0.26
Behavioural	304.84	17	351.21	19	−2.445	0.015	−0.15

M_rank_—mean of ranked value; Me—median; r_g_—Glass’s size of effect.

**Table 4 ijerph-18-07787-t004:** Assumption of Kruskal–Wallis H tests of the results of MAS-PL scores (total score, affective, cognitive, and behavioural subscales): by place of residence (N = 625).

MAS-PL (Total Score)	Rural Areas N = 102		Small Town N = 183		City N = 340		H_(2)_	*p*	ε^2^
	M_rank_	Me	M_rank_	Me	M_rank_	Me			
Global score	309.81	80	307.05	80	317.16	81	0.410	0.815	0.00
Affective	327.65	39	305.5	39	312.64	39	0.989	0.610	0.00
Cognitive	290.66	22	304.76	22	324.14	23	3.248	0.197	0.01
Behavioural	312.55	18	316.01	18	311.52	18	0.075	0.963	0.00

Post hoc analyses were not performed due to non-significant values of H tests. M_rank_—mean of ranked value; Me—median; ε^2^—size of effect.

**Table 5 ijerph-18-07787-t005:** Descriptive analysis of the MAS-PL scores (total score, affective, cognitive and behavioural subscales) by major of study.

MAS-PL (Total Score)	*N*	Global Score		Affective		Cognitive		Behavioural	
		M	Me	M	Me	M	Me	M	Me
Administration and management in healthcare	10	79.60	75.00	39.50	36.50	23.90	23.50	16.20	15.00
Biotechnology	17	79.35	79.00	36.41	35.00	23.59	23.00	19.35	19.00
Dietetics	86	77.47	77.00	38.27	38.50	21.91	22.00	17.29	17.00
Physiotherapy	139	81.12	82.00	40.06	40.00	23.05	22.00	18.00	18.00
Cosmetology	69	80.00	81.00	40.58	40.00	22.19	22.00	17.23	17.00
Nursing	44	84.55	83.00	41.16	40.50	24.27	24.00	19.11	20.00
Obstetrics	22	78.14	76.50	39.00	41.00	23.23	21.50	15.91	13.50
Emergency medical services	18	80.56	81.00	35.72	36.00	25.11	26.00	19.72	20.00
Dentistry	220	80.01	81.00	38.49	38.00	23.76	24.00	17.77	18.00

M—mean; SD—standard deviation; Me—median.

**Table 6 ijerph-18-07787-t006:** Assumption of Spearman’s correlation analysis of MAS-PL scores (total score, affective, cognitive and behavioural subscales) with age and year of study (N = 625).

MAS-PL (Total Score)	Age		Year of Study	
	r_s_	*p*	r_s_	*p*
Global score	0.00	0.915	0.01	0.734
Affective	0.00	0.926	0.07	0.068
Cognitive	0.01	0.806	−0.07	0.082
Behavioural	0.02	0.692	0.02	0.702
Age	-	-	0.70	<0.001
Year of study	0.70	<0.001	-	-

**Table 7 ijerph-18-07787-t007:** Assumption of Spearman’s correlation analysis MAS-PL scores (total score, affective, cognitive and behavioural subscales; N = 625).

MAS-PL (Total Score)	Global Score		Affective		Cognitive		Behavioural	
	r_s_	*p*	r_s_	*p*	r_s_	*p*	r_s_	*p*
Global score	--	--	0.80	<0.001	0.60	<0.001	0.66	<0.001
Affective	0.80	<0.001	--	--	0.20	<0.001	0.32	<0.001
Cognitive	0.60	<0.001	0.20	<0.001	--	--	0.25	<0.001
Behavioural	0.66	<0.001	0.32	<0.001	0.25	<0.001	--	--

**Table 8 ijerph-18-07787-t008:** Assumption of regression analysis of the results of MAS-PL scores (total score, affective, cognitive and behavioural subscales): Scheme 625.

MAS-PL (Total Score)	F_(4, 620)_	*p*	r^2^	Predictor	β	*p*
Global score	1.457	0.214	0.00	Age	−0.01	0.776
				Gender	−0.10	0.017
				Year of study	0.03	0.535
				Place of residence	0.00	0.988
Affective	1.790	0.129	0.01	Age	−0.06	0.145
				Gender	0.01	0.800
				Year of study	0.11	0.017
				Place of residence	−0.05	0.255
Cognitive	8.047	<0.001	0.04	Age	0.09	0.032
				Gender	−0.15	<0.001
				Year of study	−0.13	0.004
				Place of residence	0.08	0.044
Behavioural	2.010	0.092	0.01	Age	−0.03	0.495
				Gender	−0.11	0.006
				Year of study	0.04	0.369
				Place of residence	−0.01	0.786

## Data Availability

The data presented in this study are available on request from the corresponding author.

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
