# Peer review of "Attitudes of Polish Medical and Health Sciences Students towards Persons with Physical Disabilities Using the MAS-PL Scale"

_ijerph, 2021, doi:10.3390/ijerph18157787_

Round 1

Reviewer 1 Report

I believe that the article is suitable for publication with minor revisions.

At the outset, I would like to add that I do not comment on the editing part of the text. It is composed correctly, written clearly and legibly. The part devoted to the discussion deserves praise, where the authors refer to numerous research achievements in the subject. They also correctly diagnose the limitations caused by the method of conducting the research. However, there are a few more of them than indicated in the text. Therefore, I have to make some comments on the methodological part of the research and the results presented in the article. The text mentions that 634 students were interviewed. What technique was it made of: PAPI, CAWI? Was it an auditorium questionnaire, or was the tool a questionnaire handed out to students who brought a completed questionnaire to the next class? - it would be worth including a research tool in appendix. Where does this number of completed questionnaires come from? What was the sampling procedure: random, purposeful? How does the sample structure relate to the structure of students at the university where the study was conducted in relation to the variables presented in the article, i.e. gender, age, place of residence, major of study, year of study? Did the survey cover all fields of studies or only selected ones and why? Without this information, it is not known how reliable the obtained data is in relation to the population of universities mentioned in the text. This is evidenced, for example, by a large disproportion in the variable gender. What does it come from? Is it a result of the feminization of the studied fields of study, or were men more often refusing to participate in the survey? It is best to add a column in Table 1 containing data from the entire university.

The answer to the above questions is important in the context of the objectives of the article set by the authors:

"The aim of this study was to determine, using the MAS-PL scale:

1) Attitudes of medical students towards disabled people.

2) Differences in the attitudes towards disabled people among medical students de-pending on gender, year, major, and place of residence. "

Without describing the sampling procedure and the distribution of key variables in relation to the general population of the university or students of medicine and health sciences, we cannot extrapolate the results, which the authors themselves emphasize: "Results cannot be generalized to the rest of the medical and health sciences student population as they show attitudes of students from one medical school. " Thus, the title of the article is a bit misleading and it should be added that it concerns "Attitudes of Polish Medical and Health Sciences Students from Pomeranian Medical University in Szczecin ..."

Summing up, the article broadens the state of research on this important social issue, constituting a certain introduction to more systematic research on this topic. Therefore, they can be treated as an exploratory measurement that requires only supplementing a few methodological questions in order to be able to better interpret the obtained results and be able to relate them to the research conducted so far using the scale adopted by the authors.

Author Response

Response to Reviewer 1 Comments

Thank you very much for all Your comments - we have made the following changes before publication.

Point 1: “Therefore, I have to make some comments on the methodological part of the research and the results presented in the article. The text mentions that 634 students were interviewed. What technique was it made of: PAPI, CAWI? Was it an auditorium questionnaire, or was the tool a questionnaire handed out to students who brought a completed questionnaire to the next class? - it would be worth including a research tool in appendix. Where does this number of completed questionnaires come from? What was the sampling procedure: random, purposeful? Did the survey cover all fields of studies or only selected ones and why? This is evidenced, for example, by a large disproportion in the variable gender. What does it come from? Is it a result of the feminization of the studied fields of study, or were men more often refusing to participate in the survey?”.

Response 1: We have added more details about the research methodology (PAPI method, auditorium questionnaire, research sample) in section 2.1. Data Collection and Research Sample.

Point 2: “How does the sample structure relate to the structure of students at the university where the study was conducted in relation to the variables presented in the article, i.e. gender, age, place of residence, major of study, year of study? It is best to add a column in Table 1 containing data from the entire university”.

Response 2: We have added a column in Table 1 containing data from the entire university.

Point 3: “(…) it would be worth including a research tool in appendix”.

Response 3: The MAS-PL questionnaire was added as appendix to our previous article with open access: Radlińska, I.; Starkowska, A.; Kożybska, M.; Flaga-Gieruszyńska, K.; Karakiewicz, B. The multidimensional attitudes scale towards persons with disabilities (MAS)–a Polish adaptation (MAS-PL). Ann Agric Environ Med 2020, 27, 4, 613-620. doi: 10.26444/aaem/114531 (item 19 of the literature) and the original,  English MAS is  available in article by Findler, L.; Vilchinsky, N.; Werner, S. The multidimensional attitudes scale toward persons with disabilities (MAS) con-struction and validation. Rehabil Couns Bull 2007, 50, 3, 166-176. https://doi.org/10.1177/00343552070500030401 (item 24 of the literature).

Point 4: “Thus, the title of the article is a bit misleading and it should be added that it concerns "Attitudes of Polish Medical and Health Sciences Students from Pomeranian Medical University in Szczecin ...".

Response 4: When creating the title of our work, we modelled it on the works of other authors, e.g.  Chan, C. C. H.; Lee, T. M. C.; Yuen, H.-K.; Chan, F. Attitudes towards people with disabilities between Chinese reha-bilitation and business students: An implication for practice. Rehabil Psychol 2002, 47, 3, 324–338. https://doi.org/10.1037/0090-555 (item 47 of the literature); Sahin, H.; Akyol, A.D. Evaluation of nursing and medical students’ attitudes towards people with disabilities. J Clin Nurs 2010; 19, 2271–9 (item 50 of the literature). In our opinion, giving additional information would make the title too long - however, already by reading the abstract you can find out which university the students come from.

Reviewer 2 Report

The manuscript requires corrections before publication. They concern partly the research design and methodology, as well as the presented results and conclusions. The positive sides of the article certainly include the introduction and the discussion with extensive citations. For the aim of the study, the authors indicate the investigation on attitudes of medical students towards people with disabilities and the factors that determine them. This requires clarification, because the data analyzes concern a wider group of students (also health sciences), which is repeatedly mentioned in the article. The MAS-PL research tool used is well-established in the literature, but it seems that the authors do not indicate reference values ​​for the scale - hence it is difficult for the reader to understand what presented values ​​mean (this should be added). Moreover, it is suggested to describe the study group more precisely: what were the inclusion criteria for the study, what is the structure of the general population (students of Pomeranian Medical University in Szczecin), what are the sex and years of studies in particular fields of study (Table 1 to be completed). In the presentation of the results, since, as it was stated in , the variables being ‘scores’ did not have a normal distribution, it is necessary to take into account the median (Table 2). The results describe the lack of correlation with the MAS-score, year / field of study and age. While the year of study and age are presented in Table 4, there is no analysis of the field of study presented in table. It is also suggested to supplement the analysis with regression modeling, which would allow for multivariate analysis (taking into account the interactions between independent variables) - hence, in the discussion the authors mention "A multidimensional analysis thus reveals a hidden space (in terms of emotions) ...". Moreover, rather weak correlation between the year of studies and the age of students (0.260) is interesting – what is the reason of that? In the Limitations section, the authors mention that their results show the attitudes of students from one medical school and cannot be generalized - this sentence will be justified when we see what is the structure of the population of medical school students and how study sample reflects it. Summing up, the work can complement the literature on an important social issue.

Author Response

Response to Reviewer 2 Comments

Thank you very much for all Your comments - we have made the following changes before publication.

Point 1: “The MAS-PL research tool used is well-established in the literature, but it seems that the authors do not indicate reference values for the scale - hence it is difficult for the reader to understand what presented values mean (this should be added)”.

Response 1: Thank you for this comment, but the reference values for the scale means that higher scores represent more negative attitudes and positive items require reverse scoring – we have added this information to section 2.2. Measures. In addition we have added a brief description of the tool to better understand the results (section 2.2.) and in the results we have added an analysis of skewness and kurtosis to describe the results obtained in more detail.

 Point 2: “Moreover, it is suggested to describe the study group more precisely: what were the inclusion criteria for the study, what is the structure of the general population (students of Pomeranian Medical University in Szczecin), what are the sex and years of studies in particular fields of study (Table 1 to be completed). In the presentation of the results, since, as it was stated in , the variables being ‘scores’ did not have a normal distribution, it is necessary to take into account the median (Table 2). The results describe the lack of correlation with the MAS-score, year / field of study and age. While the year of study and age are presented in Table 4, there is no analysis of the field of study presented in table”.

Response 2: Thank you for Your accurate comments - we have made the following changes in response:

  • We have added more details about the research methodology (PAPI method, auditorium questionnaire, research sample) and a column in Table 1- the structure of the general population of students of Pomeranian Medical University in Szczecin;
  • In Table 2, we have added the median;
  • We have added Table 5 showing the results of students by field of study and we have provided the result of the Kruskal-Wallis test in the text. We did not present the test result in the table due to the very large number of variables. For the analysis of the field of study, we used the Kruskal-Wallis test and not the V-Cramer correlation due to the clearer possibility of interpreting the results. Moreover, Table 4 has been added to present the results of the Kraskal-Wallis test analysis between MAS-PL and the place of residence.

Point 3: “It is also suggested to supplement the analysis with regression modeling, which would allow for multivariate analysis (taking into account the interactions between independent variables) - hence, in the discussion the authors mention "A multidimensional analysis thus reveals a hidden space (in terms of emotions) ...” ".

Response 3: We supplemented the results with a regression analysis (see Table 8). Unfortunately, the multivariate analysis was abandoned. The number of verifiable interactions is very large, and the literature does not provide a basis for checking any of them. The study group is too small to check all interactions. When examining interactions, we would create 60 subgroups for comparison, the acceptable size for each of the subgroups would have to be 20. In our sample, we cannot provide such a size for each of the subgroups.

Point 4: “Moreover, rather weak correlation between the year of studies and the age of students (0.260) is interesting – what is the reason of that?”.

Response 4: We compared the database with questionnaires and found errors in the coding of the year of study. We have also removed the questionnaires with deficiencies from the database. For this reason, all tables in the manuscript have been changed. In the new calculations, the correlation between the age and year of study turned out to be stronger (0.70; Table 6), which is justified by the fact that higher-year students are usually older than lower-year students.

Point 5: “In the Limitations section, the authors mention that their results show the attitudes of students from one medical school and cannot be generalized - this sentence will be justified when we see what is the structure of the population of medical school students and how study sample reflects it.”

Response 5: We have added a column in Table 1- the structure of the general population of students of Pomeranian Medical University in Szczecin.

Reviewer 3 Report

I recommend  the following revisions:

  1. The introduction is well structured and has relevant content, but improvement of the scientific writing and English is suggested;
  2. It is suggested to integrate an additional short description of the instrument in this article;
  3. Explain whether inclusion and exclusion criteria were determined for participation in the study;
  4. Previous personal contact with people with disabilities seems essential for determining attitudes towards this condition. Explain why was this variable not analysed in the present study?

Reviewer 4 Report

It is a well-written and researched article about attitudes towards individuals with disabilities. Although the article does not present new idea, a particularly strong side of the article is the sample size and the importance of the topic. The abstract is missing a concluding statement. Based on the above, I suggest that the article be published with minor changes.
